# Research on Detection Methods for Gas Pipeline Networks Under Small-Hole Leakage Conditions

**DOI:** 10.3390/s25030755

**Published:** 2025-01-26

**Authors:** Ying Zhao, Lingxi Yang, Qingqing Duan, Zhiqiang Zhao, Zheng Wang

**Affiliations:** 1Bocom Intelligent Information Technology Co., Ltd., Beijing 100102, China; 2College of Intelligence and Computing, Tianjin University, Tianjin 300350, China; 3Tianjin Key Laboratory of Machine Learning, Tianjin University, Tianjin 300350, China

**Keywords:** gas pipeline networks, leak detection, STAN, deep learning

## Abstract

Gas pipeline networks are vital urban infrastructure, susceptible to leaks caused by natural disasters and adverse weather, posing significant safety risks. Detecting and localizing these leaks is crucial for mitigating hazards. However, existing methods often fail to effectively model the time-varying structural data of pipelines, limiting their detection capabilities. This study introduces a novel approach for leak detection using a spatial–temporal attention network (STAN) tailored for small-hole leakage conditions. A graph attention network (GAT) is first used to model the spatial dependencies between sensors, capturing the dynamic patterns of adjacent nodes. An LSTM model is then employed for encoding and decoding time series data, incorporating a temporal attention mechanism to capture evolving changes over time, thus improving detection accuracy. The proposed model is evaluated using Pipeline Studio software and compared with state-of-the-art models on a gas pipeline simulation dataset. Results demonstrate competitive precision (91.7%), recall (96.5%), and F1-score (0.94). Furthermore, the method effectively identifies sensor statuses and temporal dynamics, reducing leakage risks and enhancing model performance. This study highlights the potential of deep learning techniques in addressing the challenges of leak detection and emphasizes the effectiveness of spatial–temporal modeling for improved detection accuracy.

## 1. Introduction

In the context of rising global energy demand, the consumption of natural gas has also increased, positioning it as one of the world’s primary energy sources [1]. Natural gas pipelines serve as the main transport mechanism due to their low cost and high efficiency, and are widely utilized across various sectors [2]. However, as pipeline service lives extend and operating environments become more complex, pipeline leakage has become a common issue [3]. Such leaks not only result in the wastage of natural gas resources but also pose serious safety risks, threatening both human lives and the environment [4]. This risk is particularly pronounced in urban gas pipeline networks [5], highlighting the urgent need to enhance the accuracy and responsiveness of gas pipeline leakage detection technologies. Although some online monitoring systems have been implemented, they still struggle with issues such as missed detections and false alarms, largely due to external environmental changes and inherent system complexities [6]. To address these challenges, there is a pressing need to develop more intelligent and precise leakage detection technologies to ensure the safety and stability of natural gas transportation.

Leak detection and localization in gas pipeline networks aim to assess the current conditions of the pipeline system (e.g., normal or abnormal) based on historical sensor data, facilitating the detection and positioning of leaks within the gas pipeline network [7]. The working conditions of adjacent pipelines can significantly influence one another; when a leak occurs, neighboring pipelines are also affected. Furthermore, a pipeline’s operational state is intrinsically linked to its past observations.

Traditional gas leak monitoring methods can be broadly categorized into two types: physical sensor-based monitoring [8] and mathematical model-based monitoring [9]. The former relies on physical sensors to directly measure critical pipeline parameters. These sensor-based approaches can be further classified into categories such as acoustic [10,11,12,13,14], optical [15,16,17,18], and soil detection [19]. The advantages of these technologies include real-time monitoring capabilities, rapid response times, and suitability for emergency leak management. However, traditional physical sensor methods also face significant limitations, including high costs and complexities in installation and maintenance—particularly in long-distance or intricate terrain pipeline networks where achieving comprehensive coverage proves challenging. Conversely, mathematical model monitoring involves establishing a mathematical representation of the pipeline system, using deviations between actual data and model predictions to monitor pipeline conditions. While this approach excels in quantitative analysis and can provide accurate predictions regarding pipeline operations, its applicability and precision are often hindered by the complexities of pipeline network structures, particularly due to the failure to adequately account for various factors influencing fluid behavior. Thus, enhancing the real-time capabilities and applicability of traditional monitoring methods remains a critical focus of contemporary research.

With the rapid advancement of big data technologies, data-driven methodologies have increasingly gained traction in the field of gas pipeline monitoring [20]. These approaches typically involve real-time monitoring and analysis of pipeline parameters such as flow and pressure, integrated with machine learning and data mining techniques. For instance, methods like the negative pressure wave technique [21,22,23] and wavelet analysis [24,25,26] are prevalent in such monitoring systems. These advanced techniques effectively address the challenges posed by the complexity and dimensionality of sensor data, enabling the rapid identification and rectification of potential leakage issues. Nonetheless, despite their promising capabilities in gas leak detection, machine learning models must still overcome challenges such as sample complexity and limited feature interpretability. Many existing models depend heavily on supervised feature training, which, although utilizing historical data for state classification, may suffer from inadequate generalization due to their reliance on leakage samples. Researchers have increasingly recognized the advantages of deep learning technologies, particularly their capacity for adaptive feature extraction, allowing deep learning to surpass the limitations of traditional methods in processing complex signals and data [27]. Consequently, combining deep learning techniques with existing signal processing methods offers a novel solution for gas pipeline leak detection.

Current deep learning models often fail to fully exploit the potential of spatiotemporal data in analyzing gas pipeline network data, being confined to the examination of static time series or spatial features [28,29,30,31]. This narrow focus overlooks the intricate spatiotemporal correlations inherent in gas pipeline network condition detection, resulting in a limited understanding of the complexities associated with leakage events. On the one hand, dynamic spatial correlations exist within the gas pipeline network topology. As illustrated in Figure 1, the relationships between pipeline states at various sensors can change dramatically over time (for example, before and during a leak). On the other hand, dynamic temporal correlations also occur, as shown in Figure 1, where sensor data such as flow or pressure may experience significant fluctuations (for instance, at the onset of a leak), thereby influencing correlations across different time steps.

To address this issue, this study proposes an innovative network model based on temporal and spatial attention mechanisms, aimed at constructing the graph structure of the pipeline network through encoding technology. This model seeks to fully utilize the information and temporal variability of sensor distributions, thereby enhancing its adaptability to complex pipeline networks. Specifically, a graph attention network is employed to embed the features of the gas pipeline network into the graph structure, while the spatial dependencies of sensors are modeled using attention mechanisms and graph neural networks to identify the changing behaviors of adjacent nodes. Additionally, the LSTM model serves as an encoding and decoding tool for time series features, combined with the temporal attention mechanism to discern dynamic change patterns over time, further improving the model’s recognition accuracy. The primary objective of this model is to enhance the prediction accuracy of gas leaks, providing robust support for subsequent monitoring and management. Experimental results demonstrate that the model performs exceptionally well in gas pipeline leak detection, effectively capturing the characteristics of leakage events and achieving efficient monitoring and localization in complex environments.

The structure of this study is organized as follows: Section 1 provides the research background and context, along with a brief overview of the proposed methodology. Section 2 details the principles underlying the STAN model. In Section 3, the experimental setup is described, followed by an analysis of the results that demonstrate the practicality and advantages of the proposed approach. Section 4 explores the broader implications of the research findings, highlighting the challenges that must be addressed for future development. Finally, Section 5 summarizes the key conclusions drawn from the study.

## 2. Materials and Methods

Figure 2 presents the comprehensive framework of the STAN model introduced in this study. This model encompasses two primary components: a spatial attention module and a temporal attention module, which are designed to capture the spatial distribution and temporal trends of the gas network, respectively. The process begins with the encoding of the gas network’s topological structure into a graph representation. Following this, the time series dataset *X* from all sensors is embedded within this graph structure. After partitioning the data into batches, the graph representation is fed into the spatial attention module, where the graph attention mechanism identifies pertinent information across the pipelines in the network’s topology. Subsequently, the data are reformatted to reflect the temporal dimension, enabling the capture of temporal dynamics. These reformatted data are then introduced to the temporal attention module. Within this module, the data undergo encoding and decoding using Long Short-Term Memory (LSTM) networks, with the temporal dependencies effectively captured through the application of the temporal attention mechanism.

### 2.1. Sequences Input

The initial step in detecting gas pipeline leaks using the STAN model involves the construction of a multivariate time series dataset derived from sensors positioned at various locations within the pipeline network. This dataset, denoted as *X*, captures the dynamic characteristics and interactions among the sensors over time and can be represented mathematically as follows:(1)$$X={\left(\begin{array}{ccc} x_{1,1} & \cdots & x_{1,k}\\ \vdots & \ddots & \vdots \\ x_{n,1} & \cdots & x_{n,k} \end{array}\right)}_{L}$$

Here, *k* represents the number of time series signals collected by each individual sensor, *n* indicates the total number of sensors, and *L* denotes the length of the sampling period. The sequence *X* is subsequently encoded into a graph structure, which facilitates the representation of node features: (2)$$h=\left\{{\vec{h}}_{1},{\vec{h}}_{2},\cdots ,{\vec{h}}_{N}\right\},{\vec{h}}_{i}\in R^{K},$$
where *K* is the number of features associated with each node. The graph attention convolutional network layer then generates a new set of node features, potentially characterized by a different cardinality $K^{\prime }$:(3)$$h^{\prime }=\left\{{\vec{h}}_{1}^{\prime },{\vec{h}}_{2}^{\prime },\ldots ,{\vec{h}}_{N}^{\prime }\right\},{\vec{h}}_{i}^{\prime }\in R^{K^{\prime }}.$$

In a dataset of length *L*, the data are divided into multiple time batches using a sliding window of length *l*. The features within a specific batch, starting at time *t*, are expressed as follows: (4)$$X_{t:t+l}=\left\{h_{t},h_{t+1},\cdots ,h_{t+l}\right\}.$$

### 2.2. Spatial Attention Block

In the task of detecting and localizing leaks within complex gas pipeline networks, the spatial attention module plays a key role in enhancing the performance of the model. Graph Convolutional Networks (GCNs) are commonly used to analyze the spatial relationships between nodes in these networks, as they are effective at capturing dependencies among interconnected nodes—an essential capability for tasks like leak detection. However, traditional GCN methods apply uniform convolution kernels, which aggregate information from neighboring nodes to a central node without accounting for the varying influence that each neighbor may have. This uniform approach assumes that all neighboring nodes contribute equally to the central node, which is not accurate in real-world applications. In gas pipeline networks, the influence of neighboring nodes on a central node can differ significantly depending on factors such as proximity, pipe material, and environmental conditions like soil properties or weather. Given the complex topological structure of pipeline networks, which is influenced by variables like pipe length, material composition, leakage aperture, and network density, the traditional method of uniform feature aggregation can lead to inaccuracies. These factors create a heterogeneous impact on how information flows between nodes, meaning that some neighboring nodes may have a much more significant effect on the target node than others, a nuance that traditional GCNs fail to capture.

To overcome these limitations, the Graph Attention Network (GAT) introduces an attention mechanism that allows for a more nuanced approach. Unlike traditional GCNs, GAT leverages a self-attention mechanism to assign a weight to each neighboring node based on its intrinsic characteristics, rather than applying a uniform weight across all neighbors. This weighted summation approach ensures that the features extracted from each neighboring node are adjusted according to their relevance to the central node. The attention weight is dynamically computed and is independent of the overall graph structure, allowing the model to focus on the most important nodes in the vicinity of the target node. This is particularly beneficial in gas pipeline networks where the influence of a neighboring node on a central node is not constant but rather varies based on the contextual factors mentioned earlier.

This flexibility allows GAT to overcome the constraints of traditional GCNs by dynamically assigning varying learning weights to different neighborhoods, thereby enhancing the accuracy of spatial relationship modeling. Within the GAT layer, a shared linear transformation matrix *W* is first applied to each node’s feature vector, resulting in the transformed feature vector $Wh_{i}$. The attention coefficient between node *i* and its neighboring node *j* is subsequently calculated as follows: (5)$$e_{ij}=\mathrm{LeakyReLU}\left(a^{T}\left[Wh_{i}|Wh_{j}\right]\right)$$

Here, a is a learnable weight vector, [·|·] denotes the concatenation of feature vectors, and LeakyReLU acts as the activation function. To ensure comparability of the attention coefficients, they are normalized using the softmax function: (6)$$\alpha_{ij}=\frac{\exp(e_{ij})}{\sum_{r\in N_{i}}\exp\left(e_{ir}\right)}$$

In this equation, $N_{i}$ represents the set of neighboring nodes connected to node *i*. Subsequently, *F* independent attention mechanisms operate concurrently, and the results are averaged to yield the final feature representation for the node: (7)$$h_{i}^{\prime }=\sigma \left(\frac{1}{F}\sum_{f=1}^{F}\sum_{j\in N_{i}}\alpha_{ij}^{\left(f\right)}W^{\left(f\right)}h_{j}\right)$$

This multi-head attention mechanism empowers the model to capture diverse relationships among neighboring nodes while optimizing computational efficiency. It enhances feature expressiveness by enabling the focus on different aspects of the graph using distinct attention heads. Finally, the resulting data are restructured into a new temporal format to facilitate subsequent feature learning.

### 2.3. Temporal Attention Block

The sensor data from nodes within a gas pipeline network, including pressure and flow measurements, exhibit strong temporal correlations, particularly before and during leak events, when changes in pipeline flow are heavily influenced by prior conditions. To capture these temporal dependencies effectively, we employed an LSTM network, a type of recurrent neural network designed for processing sequential data. LSTMs address the vanishing gradient issue commonly associated with traditional RNNs through a gating mechanism that regulates the flow of information. Furthermore, we incorporated a temporal attention mechanism to adaptively capture relationships between nodes across different time intervals, considering specific conditions that may influence these interactions.

The temporal attention module operates within an encoder–decoder architecture that includes an LSTM encoding network, an LSTM decoding network, and a temporal attention layer. The LSTM encoder extracts temporal features from the output sequence generated by the graph convolutional layer and utilizes the temporal attention vector to pinpoint past observations that correlate strongly with the current state. Conversely, the LSTM decoder interprets the encoded temporal attention vector to detect pipelines exhibiting abnormal conditions. The LSTM model is particularly adept at processing time series data characterized by long-term dependencies, learning periodic temporal relationships, and constructing deep spatiotemporal feature representations through the integration of spatiotemporal attention vectors. The underlying mathematical principles guiding this process are illustrated in the following equations:(8)$$\begin{array}{c} \;F_{t}=\sigma \left(W_{xf}x_{t}+W_{hf}h_{t-1}+b_{f}\right)\\ \;I_{t}=\sigma \left(W_{xi}x_{t}+W_{hi}h_{t-1}+b_{i}\right)\\ \;O_{t}=\sigma \left(W_{xo}x_{t}+W_{ho}h_{t-1}+b_{o}\right)\\ \;c_{t}=F_{t}⊙c_{t-1}+I_{t}⊙\tan h\left(W_{xc}x_{t}+W_{hc}h_{t-1}+b_{c}\right)\\ \;h_{t}=O_{t}⊙\tan h\left(c_{t}\right) \end{array}$$

The output at each time step of the LSTM module is determined by the current input $x_{t}$ and the hidden state $h_{t-1}$ from the previous time step t−1. The hidden state $h_{t}$ is updated, and a new output is generated through the combined functions of the forget gate $F_{t}$, input gate $I_{t}$, and output gate $O_{t}$. At a microscopic level, the LSTM introduces a memory cell $c_{t}$, whose functionality relies on the coordinated operation of these three gates. Specifically, the forget gate discards irrelevant information, the input gate selects and stores valuable information, and the output gate determines the content to be output. A comprehensive representation of the LSTM is provided in Equation (Equation 8), while a simplified version is available in Equation (Equation 9).(9)$$h_{t},c_{t}=LSTM\left(x_{t},h_{t-1},c_{t-1}\right)$$

The architecture comprises two primary components: the encoder and the decoder. The encoder encodes the input time series data into a fixed-length vector while incorporating the hidden state from the previous time step to effectively capture temporal dependencies. The encoder’s output includes $h_{t}$ and $c_{t}$, which represent the feature information of time step *t*, as described in Equation (Equation 9). The decoder utilizes the fixed-length vector generated by the encoder to produce a predicted output sequence through the decoding process, enabling efficient time series data processing and prediction.

The decoder operates by interpreting these state vectors and recursively reinserting the results into the network to prevent rapid error propagation through prediction correction. This iterative process enhances the accuracy of leak detection within the gas pipeline network. Upon receiving the hidden and cell state vectors, $c_{t}$ and $h_{t}$, from the encoder, the decoder first utilizes these initial feature vectors to compute the subsequent states. This mechanism allows the temporal attention module to effectively focus on critical temporal features pertinent to leak incidents, thereby enabling the precise identification of compromised segments within the gas distribution system.

The temporal attention mechanism is a computational technique that analyzes temporal behavioral feature vectors by capturing the hidden state of the encoder during the generation of future time vectors [32]. Upon analyzing the task scenarios, we identified significant correlations among the sensor data of the gas pipeline network across different time periods. For instance, prior to a leak, the sensor data exhibit minor fluctuations in flow, potentially linked to environmental changes or normal equipment operation. In contrast, during a leak, the flow rate sharply increases while the pressure significantly decreases. This pronounced change effectively indicates an abnormal gas release within the pipeline. Such temporal variations can reveal the progression of leakage events. The temporal attention mechanism adeptly learns the long-term dependencies within these historical datasets, assigning higher weights to the most relevant sensor data during specific time frames. Consequently, analyzing the target weights of each time period enhances the detection of potential leakage patterns and pipeline anomalies, thereby improving the interpretability of the attention model. This approach not only increases the accuracy of gas pipeline network detection but also provides substantial support for identifying potential leakage points.(10)$$E=V_{e}\cdot \sigma \left(\left({\left(x_{i}^{\left(r-1\right)}\right)}^{T}W_{1}\right)W_{2}\left(W_{3}x_{i}^{\left(r-1\right)}\right)+b_{e}\right)$$(11)$$E_{i,j}^{\prime }=\frac{\exp\left(E_{i,j}\right)}{\sum_{j=1}^{T_{r-1}}\exp\left(E_{i,j}\right)}$$

Here, $V_{e},b_{e}\in R^{T_{r-1}\times T_{r-1}},W_{1}\in R^{N},W_{2}\in R^{C_{r-1}\times N}$, and $W_{3}\in R^{C_{r-1}}$ are learnable parameters. $t_{R-1}$ is the length of the temporal dimension in the *r*-th layer. $C_{r-1}$ is the number of channels of the input data in the *r*-th layer. The temporal attention matrix *E* is shaped by diverse inputs, with each element $E_{i,j}$ in *E* representing the semantic correlation strength between *i* and time *j*.

### 2.4. Metrics

To evaluate the performance of the STAN model in time series classification comprehensively, we employed three widely accepted metrics: precision, recall, and F1 score. These metrics effectively reflect the model’s performance in the context of gas pipeline leak detection [33,34]. Precision is defined as the ratio of true positive samples to the total number of samples classified as abnormal by the model. A high precision score indicates the model’s effectiveness in reducing false positives, thereby minimizing the misclassification of normal samples as anomalies. Conversely, recall quantifies the model’s capability to accurately identify actual anomalies, representing the proportion of true abnormal samples that are detected. It is essential to note that increasing recall often results in a trade-off with precision, as capturing more potential abnormal events may lead to more false positives. The F1 score integrates both precision and recall into a single metric, providing a balanced assessment of the model’s overall anomaly detection capability, especially in scenarios characterized by imbalanced data. Given the dynamic and complex nature of time series data, it is crucial to consider these metrics collectively when evaluating model performance. Thus, utilizing precision, recall, and F1 score offers a comprehensive understanding of the strengths and limitations of the STAN model in practical applications, informing future optimizations and refinements. The mathematical formulations for these metrics can be found in Equations (Equation 12)–(Equation 14).(12)$$Precision=\frac{TP}{TP+FP}$$(13)$$Recall=\frac{TP}{TP+FN}$$(14)$$F1=\frac{2\times Prec\times Rec}{Prec+Rec}$$

In this context, the statistics of true positives (TPs), true negatives (TNs), false positives (FPs), and false negatives (FNs) are critical for assessing model performance. TPs represent the number of abnormal samples that have been accurately identified by the model, while TNs refer to the number of normal samples that are correctly classified. Conversely, FPs indicate the number of normal samples that have been incorrectly identified as anomalies, and FNs signify the number of actual anomalies that were not detected by the model. Collectively, these metrics provide a comprehensive picture of the model’s diagnostic capability in the context of gas pipeline leak detection.

## 3. Results

### 3.1. Datasets

Due to the limited availability of publicly accessible gas leakage datasets, this study utilizes Pipeline Studio 4.2.1 to simulate the normal operation and leakage behavior of a small-scale gas pipeline network under controlled conditions [35]. Figure 3 presents the topology of the simulated natural gas pipeline network, which consists of 14 nodes and 17 pipes. These nodes represent sensors commonly deployed in real-world gas pipeline systems, each of which monitors pressure and flow signals within the network. The pipeline specifications include an inner diameter of 482.6 mm and a wall thickness of 12.7 mm. A detailed list of the input parameters is provided in Table 1. The experiment was conducted over a 24 h period in a simulated pipeline environment, using the BWRS equation, known for its broad applicability and high computational precision, as the governing equation. Constraints were applied to both the maximum flow rate at the gas source and the minimum pressure at the user endpoint. The data collection frequency aligns with typical supervisory control and data acquisition (SCADA) system parameters for urban gas transmission and distribution networks, with the natural gas composition at the gas source outlined in Table 2.

According to the classification by the European Gas Pipeline Emergency Response Data Group [36], leaks resulting from pinholes or cracks have an effective diameter of no more than 20 mm, while leaks with diameters ranging from 20 mm up to the pipe diameter are categorized as perforations. These two types of leaks are generally classified as small-hole leaks in natural gas networks and similar systems. In this study, three different leakage scenarios were created, with hole diameters of 20 mm, 40 mm, and 60 mm, each of which is significantly smaller than the pipe’s diameter of 482.6 mm. Figure 4 illustrates the distribution of six leak locations and the arrangement of 14 sensors across the network. For each leak location, three distinct scenarios were simulated, each involving a single-hole leak. These scenarios were triggered three times at different time points, with the duration of the leak, caused by the aperture, randomly set between 120 and 300 s (2 to 5 min).

The experimental setup aims to replicate real-world operating conditions, but it is limited by the absence of background noise, a key feature of actual systems. To approximate this, Gaussian noise was introduced into the simulated data and used as the raw signal. In total, 54 distinct pipeline leakage scenarios were simulated, taking into account six leakage locations, three different hole diameters, and three random leak durations. If a leak occurs within a specified sliding window, it is marked as a positive sample.

To address the issue of class imbalance between positive and negative samples, the timing of the leak onset was carefully controlled. Specifically, leaks were initiated at three different times—at the 3rd, 12th, and 21st hours of the simulation—to ensure a balanced distribution of positive and negative samples across the dataset. The random leak durations, as mentioned earlier, were applied to mitigate any bias in sample distribution, ensuring that both the positive and negative samples are represented proportionally.

### 3.2. Simulation Analysis and Processing of Gas Pipeline Network Leakage Conditions

Figure 5 and Figure 6 illustrate the transient response of sensor data from a portion of the gas network topology, specifically at leak location 2, where a 20 mm aperture leak occurs. The leak event initiates at the third hour of the simulation, rapidly develops within 5 min, and persists for a total of 24 h. The analysis of the leak pattern reveals four distinct flow clusters. Specifically, the flow variations in Pipe14 and Pipe15 exhibit the same trend and belong to the same cluster; Pipe13’s flow variations form the second cluster; Pipe7 and Pipe12’s flow variations belong to the fourth cluster; and the remaining pipelines primarily fall into the third cluster.

For the first three hours prior to the leak, the flow within the pipeline remained stable, and the system operated normally. However, upon the onset of the leak, significant changes in the flow dynamics were observed, highlighting the variation in flow between different sections of the pipeline. For instance, the flow changes in Pipe3 and Pipe4, which are in close proximity to the leak, exhibit distinctly different trends, despite their spatial closeness. This observation underscores the localized nature of the leakage event’s impact on flow behavior and emphasizes the importance of the leakage point’s position in determining flow patterns.

In contrast to the localized impact on flow, the pressure changes within the same area follow a more consistent trajectory. While leakage strongly influences local flow variations, the pressure data seem to reflect the response of the entire pipeline system, with pressure changes exhibiting a more uniform pattern. This suggests that, although pressure changes are relatively consistent, their response at different pipeline nodes may be influenced by additional factors, such as the pipeline material, distance from the leakage point, and the overall load on the pipeline system.

The distinction between flow and pressure behavior highlights the complexity of pipeline leakage dynamics. Flow data often indicate local anomalies, while pressure data provide a broader system-level response. The complementary nature of these two parameters is critical for effective leak detection and localization. Localized flow anomalies can serve as early indicators of a leak, while the system-wide pressure changes can corroborate the leak’s location. Therefore, accurate leak detection should not rely solely on a single parameter but should instead integrate both flow and pressure changes to enhance detection sensitivity and improve localization accuracy.

Significant variations often exist in the signals captured by different sensors in time series data, primarily due to differences in measurement units, such as those for pressure and flow. To mitigate these discrepancies, the min–max normalization method is employed, defined as follows:(15)$$x_{norm}=\frac{x-x_{\min}}{x_{\max}-x_{\min}}$$
where $x_{norm}$ indicates the normalized value, *x* refers to the original value, while $x_{\max}$ and $x_{\min}$ represent the maximum and minimum values of the original data, respectively. Following normalization, the experimental benchmark dataset is organized into three groups based on different leakage aperture sizes. Each group is subsequently partitioned into training, validation, and test sets using a 7:1:2 ratio.

### 3.3. Experimental Setup

This study utilizes PyTorch 1.5.1 and CUDA 10.2 to implement pipeline leakage detection. All experiments were conducted on an NVIDIA GeForce GTX 4090 to fully leverage its computational capabilities. After extensive experimental fine-tuning, the model’s hyperparameters were determined. Specifically, all graph convolution layers utilized 64 convolution filters, with a kernel size of 3 along the time dimension. The sliding window size is set to w=12, the batch size is b=32, and the shape of the model input is (b,w,n,k), where n=19 and k=2. The loss function used is cross-entropy, and the Adam optimizer is employed with an initial learning rate of $1\times 10^{-3}$. The final output matrix of the model has the shape (n,1), and leakage locations are determined by classifying the state of each sensor. For example, when a leak occurs at Leak2, the classification determines that sensors $N_{2}$ and $N_{4}$ are in a leakage state, which in turn indicates that a leak has occurred in Pipe3, as identified by these two sensors.

To ensure fairness and reproducibility, the STAN framework was benchmarked against several baseline models, including the following: (1) MLP, (2) LSTM, (3) GCN, and (4) Spatiotemporal Graph Convolutional Networks (STGCNs). These models represent state-of-the-art deep learning techniques with diverse architectures and learning strategies. Performance evaluation was conducted using metrics such as precision, recall, and F1 score, enabling a comprehensive assessment of various pipeline leakage detection methods. All experiments were performed under consistent, controlled conditions to ensure comparability. Each model was independently tested three times, and the average performance across all tests was recorded as the final result.

### 3.4. Experimental Result

Table 3 presents a comparative evaluation of various models based on precision, recall, and F1 score. The STAN model outperforms its counterparts in gas network leak detection, achieving a precision of 91.5%, a recall of 93.9%, and an F1 score of 92.7%. This high precision demonstrates STAN’s effectiveness in accurately identifying leaks while minimizing false alarms. In contrast, baseline models such as MLP (55.9%) and LSTM (70.4%) exhibit significantly lower precision, underscoring STAN’s superior performance in recognizing abnormal states. Compared to GCN (75.3%) and STGCN (83.1%), STAN not only achieves higher precision but also records a notable 4.2% improvement in recall, highlighting its advanced capability in processing dynamic data and spatial dependencies. Although GCN and STGCN perform commendably, STAN excels in handling complex spatiotemporal features, emphasizing the importance of spatiotemporal attention mechanisms in anomaly detection within gas pipeline networks.

The experimental results are illustrated in Figure 7. As the size of the leakage aperture varies, each model exhibits different degrees of improvement in accuracy (ACC). Specifically, the MLP achieves an accuracy of 54.5% at a small aperture, which increases to 62.7% as the aperture enlarges, highlighting its limitations in handling simpler scenarios. In contrast, the accuracy of LSTM model rises from 69.4% to 78.5%, demonstrating its adaptability to dynamic changes. The GCN excels in spatial feature modeling, with accuracy improving from 74.4% to 83.2%, further validating its effectiveness in managing complex data structures. The STGCN also performs robustly across different aperture sizes, particularly at larger apertures, achieving an accuracy of 91.0%. Notably, the STAN model achieves the highest accuracy, increasing from 88.0% to 97.6%, showcasing its exceptional capability to address a variety of leakage scenarios and underscoring the significance of the spatiotemporal attention mechanism in complex dynamic environments. These findings indicate that as model complexity increases, the ability to capture leakage features improves, resulting in a significant enhancement in detection accuracy.

## 4. Discussion

This paper addresses two key scientific challenges in gas pipeline leakage detection. First, the subtle nature of features associated with small-hole leaks makes accurate detection difficult. Second, the complex topology of urban gas pipeline networks complicates the analysis. Existing methods primarily rely on time series features, limiting their ability to accurately identify unknown leaks using only sensor alarm data. To address these issues, this study proposes a novel approach for monitoring gas pipeline leakage that leverages a spatiotemporal attention mechanism. The spatial attention module captures the interconnections within the network topology, while the temporal attention module extracts the serial correlations in sensor data over time, enabling the identification of critical temporal features. By integrating these mechanisms, the proposed algorithm effectively extracts spatial and temporal information, creating a unified representation. A graph attention-based feature extractor learns the network structure from a pre-constructed adjacency matrix, capturing spatial features and facilitating comprehensive spatiotemporal analysis. Additionally, the temporal attention module highlights the significance of each node over time, improving the interpretability of leakage features and enhancing leak localization accuracy.

The study demonstrates promising results for gas pipeline leakage detection using spatiotemporal attention mechanisms, though several challenges remain. First, reliance on simulated datasets may not fully represent the complexities of real-world pipeline environments, necessitating validation against diverse, realistic data. Second, the use of a pre-constructed adjacency matrix may limit adaptability to dynamic pipeline configurations, highlighting the need for adaptive graph construction techniques. Third, the computational demands of spatiotemporal attention methods require the development of efficient algorithms to reduce overhead without sacrificing accuracy. Fourth, integrating spatial and temporal attention into a cohesive framework poses additional challenges that warrant further investigation. Finally, incorporating sensor fusion techniques that combine multiple modalities, such as pressure and flow data, could provide a more comprehensive understanding of leakage events. Addressing these challenges will be essential for advancing effective and reliable monitoring solutions in gas pipeline networks.

## 5. Conclusions

This article presents a novel approach for detecting gas pipeline leaks using a spatiotemporal attention mechanism. It emphasizes the integration of spatial and temporal attention modules to enhance the accuracy and interpretability of leakage detection in complex pipeline environments. The study is supported by experimental results that demonstrate the effectiveness of the proposed method compared to various baseline models. Three main points of this article are as follows:(1)This paper introduces a spatiotemporal attention mechanism that effectively identifies interconnections in pipeline topology and captures temporal correlations in sensor data, facilitating the selection of critical features for leak detection.(2)The STAN model outperforms baseline models, achieving significant precision (91.7%), recall (96.5%), and F1 (94.0%). The findings indicate that as the size of the leakage aperture varies, each model exhibits different levels of accuracy improvement, with the STAN model demonstrating superior performance.(3)This article discusses the limitations of current methods, such as reliance on simulated datasets and challenges in adapting to dynamic pipeline environments. It emphasizes the need for further research to optimize the proposed model and explore sensor fusion techniques to enhance detection accuracy.

## Figures and Tables

**Figure 1 sensors-25-00755-f001:**
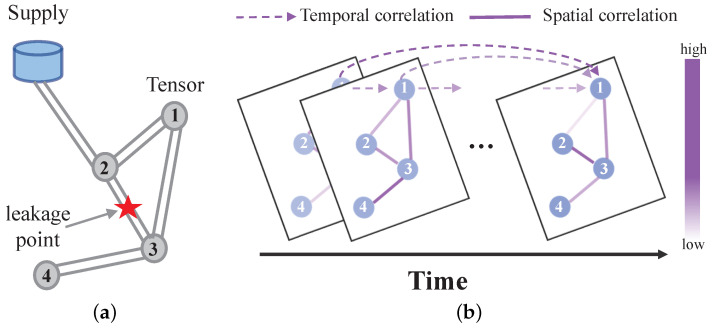
Complex spatiotemporal correlation. (**a**) Gas network structure: There are four sensors and one leak point, and supply is represented by the gas supply station. (**b**) Dynamic spatial correlation: Adjacent sensors are not always highly correlated. For example, the correlation between sensors 1 and 2 in the figure weakens over time. Dynamic temporal correlation: The network flow represented by the current sensor may be more correlated with the flow when the leak occurs at a distant time.

**Figure 2 sensors-25-00755-f002:**
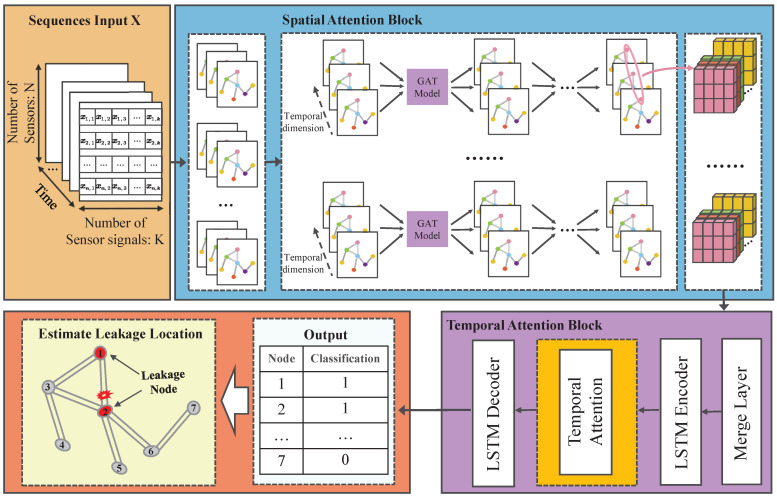
Architecture of the proposed STAN method.

**Figure 3 sensors-25-00755-f003:**
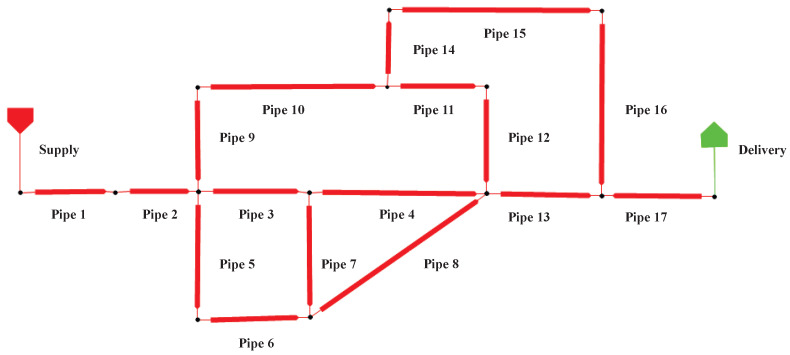
Gas network leakage model for experiments.

**Figure 4 sensors-25-00755-f004:**
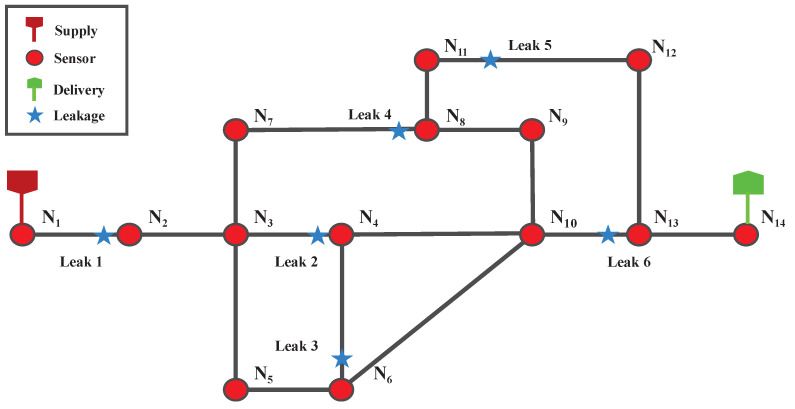
Gas network leakage model for experiments.

**Figure 5 sensors-25-00755-f005:**
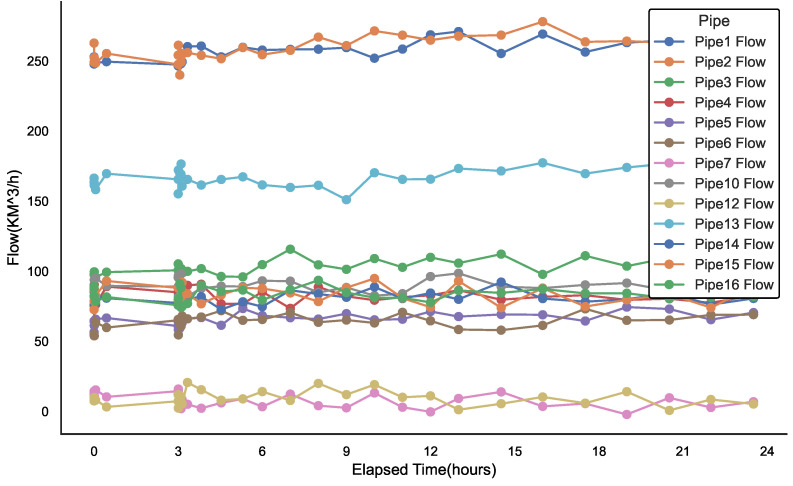
Transient operating conditions of each pipeline downstream flow rate with a 20 mm leakage aperture in the pipe network structure.

**Figure 6 sensors-25-00755-f006:**
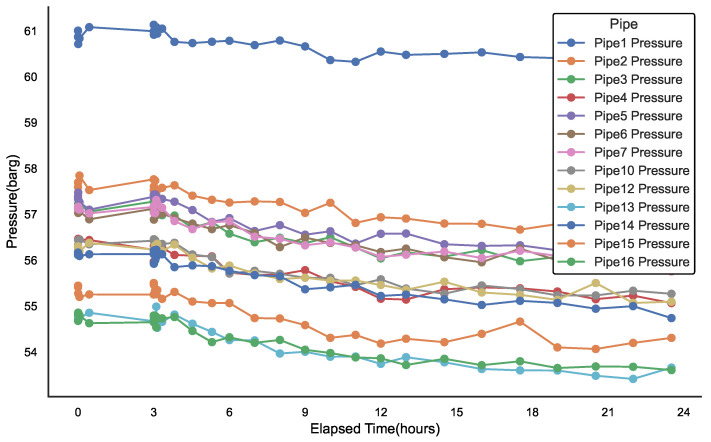
Transient working conditions of each pipeline pressure with a 20 mm leakage hole in the pipe network structure.

**Figure 7 sensors-25-00755-f007:**
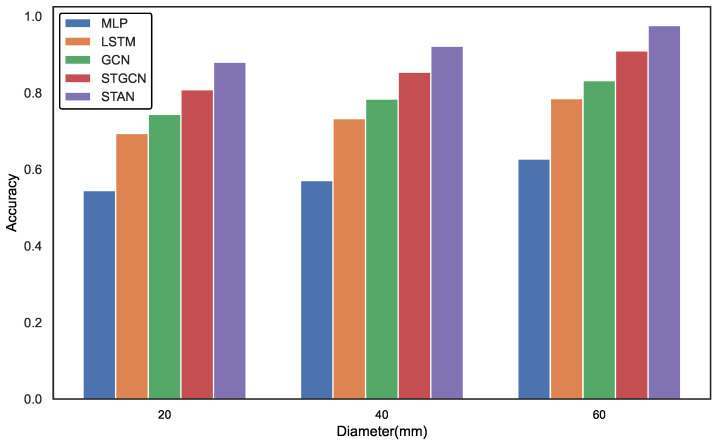
Comparison of the performance of different models in terms of accuracy under the conditions of leak aperture of 20 mm, 40 mm, and 60 mm.

**Table 1 sensors-25-00755-t001:** Pipeline parameters panel in the gas network leakage model.

Name	Length (km)	Diameter (cm)	Wall Thickness (mm)	Roughness (Micron)	Knot Spacing (km)	Efficiency
Pipe1	20	48.26	12.7	25.4	1.60934	1
Pipe2	20	48.26	12.7	25.4	1.60934	1
Pipe3	20	48.26	12.7	25.4	1.60934	1
Pipe4	40	48.26	12.7	25.4	1.60934	1
Pipe5	30	48.26	12.7	25.4	1.60934	1
Pipe6	20	48.26	12.7	25.4	1.60934	1
Pipe7	30	48.26	12.7	25.4	1.60934	1
Pipe8	50	48.26	12.7	25.4	1.60934	1
Pipe9	20	48.26	12.7	25.4	1.60934	1
Pipe10	40	48.26	12.7	25.4	1.60934	1
Pipe11	20	48.26	12.7	25.4	1.60934	1
Pipe12	20	48.26	12.7	25.4	1.60934	1
Pipe13	30	48.26	12.7	25.4	1.60934	1
Pipe14	10	48.26	12.7	25.4	1.60934	1
Pipe15	40	48.26	12.7	25.4	1.60934	1
Pipe16	30	48.26	12.7	25.4	1.60934	1
Pipe17	20	48.26	12.7	25.4	1.60934	1

**Table 2 sensors-25-00755-t002:** Natural gas composition.

Ingredient	C1	C2	C3	IC4	NC4	IC5	NC5	C6	CO_2_	N_2_	H_2_O
**mol%**	99.591	0.078	0.021	0.01	0.002	0.001	0.001	0.001	0.052	0.223	0.02

**Table 3 sensors-25-00755-t003:** Performance comparison of different models in terms of precision, recall, and F1.

Model	Precision (%)	Recall (%)	F1
MLP	55.9	77.2	64.8
LSTM	70.4	81.9	75.7
GCN	75.3	85.3	80.0
STGCN	83.1	89.7	86.3
STAN	91.5	93.9	92.7

## Data Availability

The data used to support the findings of this study are available form the corresponding author upon request.

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
