# Peer review of "Research on Detection Methods for Gas Pipeline Networks Under Small-Hole Leakage Conditions"

_sensors, 2025, doi:10.3390/s25030755_

Round 1
Reviewer 1 Report
Comments and Suggestions for Authors
Dear authors,
Congratulations on the paper, it is relevant, deals with a commom problem with not many solutions.
However, my main concern is related to the applicability of the model, and the lack of data to support it, ans also, a comparison with "sensor-based" results and other mathematical (physics-based) solutions.
I understand that you are interested in comparing different ML algorithms, however, there are other physics model, for example, based on nodal analysis that can provide very good results.
Also, isn't the amount of data really small for comparing ML-based solutions?
Also, as you simulated an experiment, maybe you have good results of the sensors, couls you include a sensor-based solution in the comparison?
Author Response
Comments 1: About the applicability of the model, the lack of sufficient data, and the need for a comparison with sensor-based and physics-based solutions, such as nodal analysis.
Response 1: Thank you for pointing this out. We agree with this comment. Therefore, We have addressed these points in the revised manuscript as follows:
On the Applicability of the Model and Mathematical (Physics-Based) Solutions: In our study, we redesigned a more realistic gas network topology and introduced noise into the simulated data to approximate sensor-based data (In chapter. 3). This approach allows for a more accurate representation of real-world scenarios. We acknowledge that while nodal analysis methods perform well in simple, noise-free datasets, their effectiveness diminishes when applied to complex topologies with noise. Specifically, the limitations of Mathematical Solutions are as follows:
- Manual Threshold Setting: Nodal analysis requires the manual setting of thresholds, which can lead to suboptimal experimental results.
- Dependency on Empirical Physical Formulas: It relies heavily on experience-based physical formulas, which limits its applicability to complex systems.
Article [1] highlights the limitations of physical methods, pointing out that these methods struggle to scale with large datasets and complex topologies. As such, future work should explore the application of machine learning approaches to these challenging scenarios. While nodal analysis has advantages in idealized environments (e.g., simple networks with no noise), machine learning methods are better suited to handle the variability and complexity inherent in real-world data. In this context, we propose a method for detecting gas pipeline leaks in industrial settings.
On the Size of the Dataset: We have provided further clarification regarding the size of the dataset, the positive-negative sample ratio, and other relevant details (subsection 3.1). While historical leakage data are indeed limited, we refer to the work in [2], which presents a method for generating leakage samples using Pipeline Studio, proving that simulated data can accurately reflect real leakage scenarios. Therefore, we performed experiments based on the simulated data, ensuring the reliability and authenticity of the results.
On the Comparison with Sensor-Based Data: In our experimental setup, we have included a comparison with sensor-based results. Specifically, we compared the performance of the machine learning model against sensor data in the noisy simulated environment. This comparison has been added in the revised manuscript (subsection 3.2~3.4). While sensor-based solutions are effective in certain cases, machine learning methods exhibit greater adaptability and robustness, particularly in complex leakage scenarios and large-scale networks.
[1]. Korlapati, N.V.S.; Khan, F.; Noor, Q.; Mirza, S.; Vaddiraju, S. Review and analysis of pipeline leak detection methods. Journal of pipeline science and engineering 2022, 2, 100074.
[2]. Gong, X.; Liu, L.; Ma, L.; Dai, J.; Zhang, H.; Liang, J.; Liang, S. A leak sample dataset construction method for gas pipeline leakage estimation using pipeline studio. In Proceedings of the 2021 International Conference on Advanced Mechatronic Systems (ICAMechS). IEEE, 2021, pp. 28–32.

Reviewer 2 Report
Comments and Suggestions for Authors
This paper dealt with the leak detection method for gas pipeline nwtworks using STAN. The proposed model is resonable for the application, but some modification must be needed for the puplication.
1. Line 38~43 in Chap 1. Introduction should be removed. It's redundant.
2. The notation in Eq. (4) should be corrected.
3. The structure of the STAN network should be presented in Chap. 3. The input matrix size, output matrix size, filter size, number of parameters, and so on should be provided. In Chapter 3, describe in detail how you applied your model to your problem.
4. Fig. 4 and 5 is the simulation result. Please described how to construct the input data.
5. Looking at figs. 4 and 5, it is clear if there is a leak and where it is located. Why use such a complex deep learning approach to this simple problem? It seems like an overly simplistic problem.
Author Response
Comments 1: Line 38~43 in Chap 1. Introduction should be removed. It's redundant.
Response 1: Thank you for pointing this out. We agreewith this comment. Therefore, We have carefully reviewed the suggested lines (38–43) in Chapter 1. Introduction. In response to the reviewer’s comment, we have removed these lines as they were deemed redundant, and this revision helps to streamline the introduction.
Comments 2: The notation in Eq. (4) should be corrected.
Response 2: Thank you for pointing this out. We agreewith this comment. Therefore,We have carefully examined the notation and made the necessary corrections to ensure consistency and clarity. The revised equation is now presented with the correct notation.
Comments 3: The structure of the STAN network should be presented in Chap. 3. The input matrix size, output matrix size, filter size, number of parameters, and so on should be provided. In Chapter 3, describe in detail how you applied your model to your problem.
Response 3: Thank you for pointing this out. We agreewith this comment. Therefore,we have added a detailed description of the STAN network structure in the subsection 3.3 Experimental Setup(lines 358~369). This includes the input matrix size, output matrix size, filter size, number of parameters, and other relevant hyperparameters. Additionally, we have elaborated on how the model was applied to the specific problem, providing a clearer understanding of the implementation and experimental setup.
Comments 4: Fig. 4 and 5 is the simulation result. Please described how to construct the input data.
Response 4: Thank you for pointing this out. We agreewith this comment. Therefore, we have added a description of how the dataset was constructed in the subsection 3.1 Datasets(lines 277~315). Additionally, we have explained the preprocessing steps applied to the data and how it was transformed into the input shape required by the model. The details of the input data shape and its preparation are now provided in the subsection 3.3 Experimental Setup(lines 358~369).
Comments 5: Looking at figs. 4 and 5, it is clear if there is a leak and where it is located. Why use such a complex deep learning approach to this simple problem? It seems like an overly simplistic problem.
Response 5: Thank you for pointing this out. We agreewith this comment. Therefore, We have considered the point raised regarding the simplicity of the problem. The large numerical discrepancies between pipelines in Figures 4 and 5 tend to obscure the temporal trends of the time-series data, making the leak detection task appear straightforward. However, the real challenge lies in detecting leaks in more complex scenarios, where multiple variables and noise can affect the signal. To further validate the applicability of our model, we have updated the experimental topology as shown in Figure 3. Figure 4 highlights the distribution of leak locations, while Figures 5 and 6 present the corresponding simulation results. The experimental outcomes are provided in Table 3 and Figure 7, which demonstrate the robustness of the proposed approach in handling more challenging scenarios.

Reviewer 3 Report
Comments and Suggestions for Authors
This study highlights the potential of spatial-temporal attention modeling in improving leak detection performance. The topic is engaging, and the method is innovative, making the work a noteworthy contribution to the field. However, several experimental aspects require clarification and refinement.
1、The dataset composition, size, and positive-negative sample ratio are unclear. Additionally, the training conditions, including hyperparameters, are insufficiently explained, which affects the reliability of the results.
2、STGCN's performance being worse than both LSTM and GCN under the 20mm leakage condition is unusual, indicating potential issues with training or the dataset. Further explanation and supporting evidence are required.
3、Figures 4 and 5: Both figures have the same flaw: there are seven curves in the legend but only six in the diagrams. Additionally, the legends obscure part of the curves. These issues need to be resolved.
4、Lines 316 to 319 imply that pressure and flow data are mixed in the same datasets. Please clarify the composition of the datasets.
5、The dataset description is vague. Please provide details about the size of the datasets and the positive-negative sample ratio.
6、Clarify the training conditions, including the provision of hyperparameters, as these details are necessary for reproducibility.
7、The lack of information about the training process makes the results unreliable. Please provide a detailed explanation.
8、The results lack convincing evidence due to the absence of an explanation for training randomness. Address this to improve reliability.
9、Figure 6: Under the 20mm leakage condition, STGCN’s accuracy is worse than LSTM and GCN. This is an anomaly. If GCN outperforms STGCN, STGCN should at least outperform LSTM. This discrepancy suggests potential issues with the training process or datasets. Please provide an explanation and supporting evidence.
Author Response
Comments 1: The dataset composition, size, and positive-negative sample ratio are unclear. Additionally, the training conditions, including hyperparameters, are insufficiently explained, which affects the reliability of the results.
Response 1: Thank you for pointing this out. We agreewith this comment. Therefore, we have updated the manuscript to provide a clearer description of the dataset composition, size, and the positive-negative sample ratio, which can now be found in the subsection 3.1 Datasets(lines 277~315). Additionally, we have expanded on the training conditions, including the hyperparameters used, which are detailed in the subsection 3.3 Experimental Setup(lines 358~369). These additions aim to enhance the transparency and reliability of the results.
Comments 2: STGCN's performance being worse than both LSTM and GCN under the 20mm leakage condition is unusual, indicating potential issues with training or the dataset. Further explanation and supporting evidence are required.
Response 2: Thank you for pointing this out. We agreewith this comment. Therefore, We have carefully considered the reviewer’s comment regarding the unusual performance of STGCN being worse than both LSTM and GCN under the 20mm leakage condition. Upon further analysis, we found that this result can be explained by the specific characteristics of the simulation environment. STGCN, which was originally proposed for spatiotemporal traffic flow prediction, performs well when there are clear peak patterns, such as those found in traffic data. In our case, the absence of noise and the shorter steady-state time for smaller leakage apertures (e.g., 20mm) led to less pronounced features for the model to capture, resulting in suboptimal performance for smaller leakages. However, as the leakage aperture increases, the time to reach steady-state becomes longer, allowing STGCN to better capture the dynamics of the system. This is reflected in the improved performance of STGCN in experiments with larger leakage apertures, where it outperforms both LSTM and GCN. To further validate the model's performance, we have conducted additional experiments with a more realistic gas pipeline topology, as shown in Figure 3, and added noise to the simulation data to better represent real-world scenarios. The updated results, including the distribution of leak locations in Figure 4, the simulation results in Figures 5 and 6, and the corresponding experimental outcomes in Table 3 and Figure 7, provide more accurate insights into the model's effectiveness.
Comments 3: Figures 4 and 5: Both figures have the same flaw: there are seven curves in the legend but only six in the diagrams. Additionally, the legends obscure part of the curves. These issues need to be resolved.
Response 3: Thank you for pointing this out. We agreewith this comment. Therefore, We have addressed the issues mentioned: the mismatch between the number of curves in the legend and the diagrams, as well as the overlapping legends that obscure part of the curves. These issues have been resolved in the updated simulation figures, which are now presented as Figures 5 and 6.
Comments 4: Lines 316 to 319 imply that pressure and flow data are mixed in the same datasets. Please clarify the composition of the datasets.
Response 4: Thank you for pointing this out. We agreewith this comment. Therefore, We have clarified the composition of the datasets in response to the reviewer’s concern. In Subsection 2.1. Sequences Input, we introduced k to represent the types of sensor signals. Specifically, in lines 281~282, we describe how the sensors record both pressure and flow signals. Additionally, the Experimental Setup section explains the input shape of the model. The dataset construction process, including how these signals are organized, is detailed in lines 358~369. Furthermore, we verified the importance of flow and pressure for leak detection in the analysis of simulation data(lines 340~347).
Comments 5: The dataset description is vague. Please provide details about the size of the datasets and the positive-negative sample ratio.
Response 5: Thank you for pointing this out. We agreewith this comment. Therefore, We have provided a more comprehensive explanation of the dataset size and the positive-negative sample ratio. These details can now be found in the manuscript in the subsection 3.1 Datasets(lines 277~315).
Comments 6: Clarify the training conditions, including the provision of hyperparameters, as these details are necessary for reproducibility.
Response 6: Thank you for pointing this out. We agreewith this comment. Therefore, We have added the relevant details about the training conditions, including the hyperparameters used, in the manuscript. These can now be found in the subsection 3.3 Experimental Setup(lines 358~369).
Comments 7: The lack of information about the training process makes the results unreliable. Please provide a detailed explanation.
Response 7: Thank you for pointing this out. We agreewith this comment. Therefore, We have expanded the manuscript to include a thorough description of the training process, including the training procedure, optimization methods, loss functions, and evaluation metrics. These details are now provided in the subsection 3.3 Experimental Setup(lines 358~378). We hope this additional information enhances the reproducibility and clarity of the reported results.
Comments 8: The results lack convincing evidence due to the absence of an explanation for training randomness. Address this to improve reliability.
Response 8: Thank you for pointing this out. We agreewith this comment. Therefore, We have acknowledged the concern regarding the reliability of the results due to the random initialization of parameters in neural networks. To address this, we have conducted three independent experiments and reported the average of the results to mitigate any variability. Line 376~378.
Comments 9: Figure 6: Under the 20mm leakage condition, STGCN’s accuracy is worse than LSTM and GCN. This is an anomaly. If GCN outperforms STGCN, STGCN should at least outperform LSTM. This discrepancy suggests potential issues with the training process or datasets. Please provide an explanation and supporting evidence.

Round 2
Reviewer 2 Report
Comments and Suggestions for Authors
The comments have been appropriately reflected. I agree to the publication of this paper.
Reviewer 3 Report
Comments and Suggestions for Authors
The authors have provided thoughtful and detailed responses to the reviewers' comments, and the revisions have noticeably improved the manuscript's clarity and scientific rigor. While further refinements of language details may still be considered, the current version appears to be a valuable contribution to the field and could be considered for publication.